

# Assessing the cover crop effect on soil hydraulic properties by inverse modelling in a 10-year field trial

Jose Luis Gabriel[1], Miguel Quemada[2], Diana Martín-Lammerding[1], Marnik Vanclooster[3]

[1] Dpto. Medio Ambiente, INIA-INAGEA (National Institute for Agricultural and Food Research and Technology), Ctra. de la Coruña km 7.5, 28040 Madrid, Spain
[2] Dpto. Producción Agraria, Universidad Politécnica de Madrid, Avda. Puerta de Hierro 2-4,28040 Madrid, Spain.
[3] Earth and Life Institute, Université catholique de Louvain, Croix du Sud 2, B-1348 Louvain-la-Neuve, Belgium.

*Correspondence to*: Jose L. Gabriel (gabriel.jose@inia.es)

**Abstract.** Cover cropping in agriculture is expected to enhance many agricultural and ecosystems functions and services. Yet, few studies are available allowing to evaluate the impact of cover cropping on the long term change of soil hydrologic functions. We assessed the long term change of the soil hydraulic properties due to cover cropping by means of a 10-year field experiment. We monitored continuously soil water content in non cover cropped and cover cropped fields by means of capacitance probes. We subsequently determined the hydraulic properties by inverting the soil hydrological model WAVE, using the time series of the 10 year monitoring data in the object function. We observed two main impacts, each having their own time dynamics. First, we observed an initial compaction as a result of the minimum tillage. This initial negative effect was followed by a more positive cover crop effect. The positive cover crop effect consisted in an increase of the soil micro- and macro-porosity, improving the structure. This resulted in a larger soil water retention capacity. This latter improvement was mainly observed below 20 cm, and mostly in the soil layer between 40 and 80 cm depth. This study shows that the expected cover crop competition for water with the main crop can be compensated by an improvement of the water retention in the intermediate layers of the soil profile. This may enhance the hydrologic functions of agricultural soils in arid and semiarid regions which often are constrained by water stress.

## 1 Introduction

Water availability in the soil root zone is a critical agriculture production factor in semiarid and arid conditions. This water availability is mainly determined by the field scale soil water balance and controlled by hydraulic properties in the soil root zone. The properties of the soil pores, such as soil pore size distribution, connectivity and tortuosity, are directly related to several soil processes, such as soil water infiltration, retention and drainage. Macropores (500-50 µm) are related to water and air movement, whereas mesopores (50-5 µm) are related to soil water retention capacity (Carter and Ball, 1993). Both abiotic (e.g., tillage, drying and wetting) and biotic factors (e.g., roots, earthworms) influence the size, shape and continuity of the soil pores that affect the soil hydraulic processes and in turn the field water balance (Kay and VandenBygaart, 2002). Soil management can control these factors (Van Es et al., 1999), including the soil root zone water availability.



Introducing conservation practices such as minimum tillage after the long-term traditional use of mouldboard ploughing is now well known to induce changes in soil properties (Strudley et al., 2008). Initially, an increase in soil bulk density may occur (Moret and Arrúe, 2007; Rücknagel et al., 2016). However, in the long term, organic matter and aggregate stability increase, soil size pore distribution changes and, often, the soil water retention capacity increases (Strudley et al., 2008). Moreover,

Hargrove (1991) observed that plant cover helped to reduce soil compaction and soil erosion under no-tillage conditions. Cover crops (CCs) have been recognized as a successful method for improving nutrient availability (Thorup-Kristensen et al., 2003), controlling nitrate leaching (McCracken et al., 1994), reducing weed infestation (Leavitt et al., 2011) or disease suppression (Abawi and Widmer, 2000). But CC have been also recognized for soil quality improvement (Kuo et al., 1997) and soil erosion control (Langdale, 1991; Bowman et al., 2000). Leaving the soil fallow will increase soil erodibility and bulk density, which

is critical for runoff initiation (Cerdà et al., 2009). Maintaining a CC between main summer crops, when the majority of the yearly rainfall occurs under Mediterranean conditions, protects soil from the impact of rainfall, reducing aggregate slaking and disruption (i.e., crusting). Moreover, CC residues after the termination date provide a mulch on the soil surface that may prevent the direct loss of soil water through evaporation and hence increase soil root zone water availability. These benefits may counteract the possible competition for water with the main crop, which is one of the main drawbacks of CCs application

(Unger and Vigil, 1998).

During the last few years, several publications have tried to quantify the effects of CCs on soil properties (Celette et al., 2008; Ward et al., 2012; Gabriel et al., 2014). However, given the high complexity of the CC-main crop-soil-weather system, those studies do not often reach the same conclusions. Most of the publications focused on the increase in soil organic matter and its related effect on aggregate stability and soil structure (Peregrina et al., 2010; García-González et al., 2016; Rorick and

Kladivko, 2017). Other studies focused on soil hydraulic properties. For example, Celette et al. (2008) measured an increase in soil water infiltration under CC and also observed an enhancement in soil aggregate stability, whereas Quemada and Cabrera (2002), Ward et al. (2012) or Basche et al. (2016) reported an increase in soil water holding capacity when CCs were used. However, most of these studies elucidated impacts of CCs either on the soil surface or in the short term. Thus, there is a need to study the effects of CCs on soil hydraulic properties in the long term and along the soil profile. This effect is critical for

evaluating the sustainability of CCs in agro-ecosystems under current and future climate conditions, even more when the availability of water is not assured and models are predicting less rainfall in Mediterranean regions (Kaye and Quemada, 2017). Numerical models based on physical equations and knowledge of the soil hydraulic properties allow the quantification of water fluxes and water balance state variables in the soil-crop-atmosphere continuum, including water drainage (Muñoz-Carpena et al., 2008). However, complex models (simulating soil-water-plant continuum) involve a large number of parameters, with the

final prediction success relying on accurate parameter identification and on the model's sensitivity to these parameters (Šimůnek et al., 1999). Inverse modelling can be used to overcome this parameter limitation problem, allowing the identification of soil hydraulic parameters. The process consists of a search for the best set of parameters, varying the parameters iteratively and comparing the numerical prediction of the soil water state variable provided by the model with the actual value measured under field conditions (Šimůnek et al., 1999). Consequently, the main advantage of this procedure is



that the results are based on directly monitored variables (Ritter et al., 2003). In this study, the approach is made possible owing to the development of multisensor probes, which allow the continuous monitoring of soil water content at different depths with minimum soil disturbance (Fares and Alva, 2000). Inverse modelling allows the identification of parameters of the soil-crop system that are consistent with monitored soil water and crop parameters in the field and will ultimately result in

lower model prediction uncertainties. If soil-crop management dynamically affects the soil hydraulic properties and hence the soil water dynamics, then inverse modelling should enable the dynamics of soil hydraulic properties to be unravelled based on the monitoring of soil water dynamics.

The main objective of this study was to assess the medium-term effect of cover crops on soil hydraulic properties over the soil profile using the inverse modelling of a mechanistic model based on continuous soil moisture measurements in a field

experiment.

## 2 Material and Methods

### 2.1 Field experimental setup

The study was conducted in an experimental field station located in the Tajo River Basin (40º03'N, 03º31'W and 550 m a.s.l., in Aranjuez, close to Madrid, Spain). The soil was classified as a *Typic Calcixerept* based on the Soil Survey Staff (2014)

classification. Gabriel et al. (2010) defined it in 2006, at the beginning of the experiment, as a silty clay loam texture, and defined different soil chemical and physical parameters to 1.2 m in depth. Climatic conditions based on Köppen-Geiger classification (Peel et al., 2007) were defined as cold steppe arid, and the most important weather information (temperature, humidity, wind speed, precipitation and solar radiation) were recorded hourly using a field micrologger Campbell Scientific weather station (CR23X, Campbell Scientific, Logan, UT, USA) placed in the experimental field during the study period.

The field experiment consisted of a 10-year crop rotation (from October 2006 to September 2016), with or without a winter CC between consecutive main summer crops. The main crops were sown during April and harvested around September and they were maize (*Zea mays* L.; the summers of 2007, 2008, 2009, 2010, 2013, 2014 and 2016) and sunflower (*Helianthus annuus* L.; the summers of 2012 and 2015). In the summer of 2011, the field was fallow to control weeds and finished with a maize monoculture. The experimental plots consisted of eight square 144 m² plots randomly distributed between the two

treatments with four replications. Each plot received the same treatment throughout the 10-year period. The CC was barley (*Hordeum vulgare* L.) and was sown every year during the first half of October and killed during the second half of March. Barley was sown by broadcasting, incorporated with a 5-cm shallow cultivator, terminated with one application of glyphosate and chopped before main crop sowing, leaving the residues over the ground. The main crops were directly sown over the CC residues. Then, the soil was not tilled except for the first 5 cm once per year. Other cropping techniques applied to the main

crops, such as fertilization, irrigation, or weed control, were equal in both treatments and adjusted to crop demands or weed infestation. More details can be found in García-González et al. (2016).



## 2.2 Field measurements

The soil water content was monitored daily (averaging hourly measurements) using EnviroSCAN® capacitance probes (Paltineanu and Starr, 1997). Six access tubes were installed in the 10-year experiment, three per treatment. Each access tube consisted of a plastic extrusion with 6 sensors from 0.1 to 1.1 m in depth every 0.2 m. Sensors were previously normalized,

calibrated (under field and laboratory conditions) and validated following Gabriel et al. (2010).

Different soil properties were measured at the beginning of the experiment. Four layers were defined (0-20, 20-40, 40-80 and 80-120 cm depths) based on the soil description in two trenches dug on the sides of the experimental field in 2006 (Gabriel and Quemada, 2011). Ten 100 cm$^3$ (50 mm in diameter) undisturbed soil cores for each layer were taken in 2006. The soil hydraulic conductivity ($K_s$) was measured using a laboratory constant head permeameter (Klute and Dirksen, 1986), and the

saturation soil water content ($\theta_s$) was obtained as the porosity measured in the soil cores. The residual soil water content ($\theta_r$) was obtained from the lowest water content observed in each EnviroSCAN® probe after dry summer periods. The van Genuchten curve-shape initial hydraulic parameters $\alpha$ and $n$ (van Genuchten, 1980; Mualem, 1976) were obtained from the adjustment of the soil water content data observed at saturation, field capacity (Assouline and Or, 2014) and $\theta_r$ for each core and probe. These results provided a range in which soil hydraulic property values estimated by inverse modelling should be

included.

Climatic conditions (temperature, humidity, radiation, photosynthetically active radiation, rainfall and wind) were measured by a weather station located < 100 m from the field trial. Cover crop biomass was measured just before glyphosate application. Four 0.5 m × 0.5 m squares were randomly harvested from each plot, cut by hand at ground level, oven-dried at 65 °C and weighed as dry matter (d.m.). Soil cover was monitored every 15 days by taking digital images in 5 permanent points per plot

that were analysed using techniques following Ramirez-Garcia et al. (2012). The root depth was estimated based on the differential water extraction between day and night at each depth observed in the EnviroSCAN® hourly measurements, following Gabriel et al. (2012).

## 2.3 WAVE model

The Water and Agrochemicals in soil, crop and Vadose Environment (WAVE_ model (Vanclooster et al., 1996) describes the

water flow and solute movement in the vadose zone. The model numerically solves the one-dimensional isothermal Richard's equation parameterized with the van Genuchten (1980) water-retention curve and the Mualem (1976) unsaturated hydraulic conductivity. More details can be found in Gabriel et al. (2012). In this case, a Matlab® (The MathWorks Inc., Natick, MA, USA) version of the model was used. A crop subroutine was also included, principally based on the WOFOST crop model (van Diepen et al., 1989) and the previous SUCROS code (van Keulen, 1982; Spitters et al., 1988).

The climatic input data of the model are temperature and photosynthetically active radiation (assuming equal to 50% of the total radiation). Radiation is converted to an increase in biomass (based on light interception, maintenance rate and conversion efficiency) and an increase in biomass in leaf area index (LAI) (based on partitioning coefficients and leaf morphology



coefficients). The subroutine discriminates assimilate partitioning at different phenological stages: from sowing to emergence (phase 0), from emergence to flowering (vegetative phase or phase 1) and from flowering to the dead leaves phase (reproductive phase or phase 2). The phenological stage and the simulated crop height were both based on thermal time. The model considers a bounded temperature range in which thermal time is accumulated. Crop growth is also corrected by temperature, water or nitrogen stresses, reducing crop growth and increasing the senescence rate. Root development considers a triangular distribution of root biomass by depth, increasing the depth based on the thermal time until a maximum root depth. The crop subroutine parameters were calibrated prior to the identification of soil hydraulic properties using observed crop data from the first year of the experiment and validated with data from the remaining years. Phenology development rate parameters were manually adjusted first to the observed phenology in the field. Subsequently, growth parameters (LAI, aerial biomass, root depth and root biomass) were adjusted to the crop growth observations in the field.

Previous versions of the WAVE model included potential evapotranspiration as a climatic input. In this case, a subroutine was included to calculate the potential evapotranspiration described by Allen et al. (1998), based on the dual coefficient. This dual coefficient considers two coefficients multiplying the reference evapotranspiration (instead of just a single coefficient), one for evaporation and the other for transpiration, considering also the percentage of soil covered by residues or plants. Evaporation only occurs on the surface that is not covered with crop biomass, whereas transpiration only occurs on the surface covered with crop biomass. Three scenarios were considered: i) soil with crops growing and transpiring; ii) soil without crops; and iii) soil covered with dead crop residues. When the crops are transpiring, evaporation and transpiration are estimated independently. The evaporation is the result of the multiplication of the reference evapotranspiration by 1 (as the FAO coefficient suggestion for bare soil) and by the proportion of soil not covered by crops. The transpiration is the result of the multiplication of the reference evapotranspiration by the basic crop coefficient following the plateau model described by Allen et al. (1998). This model uses an initial value (for soils covered by less than 10%) for the basic crop coefficient, followed by a linear increase until a maximum value is reached when the crop covers more than 80% of the total surface. After this plateau, there is a linear decrease of the basic crop coefficient until a final value at harvest time is reached. When the soil is not covered by any crop or residue, the transpiration is equal to 0 and the evaporation is equal to the reference evapotranspiration. When the soil is covered by some dead residue, the transpiration is equal to 0 and the evaporation is calculated for the bare soil but multiplied by the fraction of soil that is not covered. The soil cover fraction was obtained from the LAI simulated and corrected following Ramirez-Garcia et al. (2012).

## 2.4 Soil hydraulic parameter identification

The inverse calibration of the soil hydraulic parameters was done based on the daily soil water content measurements. The following soil hydraulic parameters were considered to be affected by CCs and identified with a Monte Carlo based inverse modelling procedure: Ks (cm day$^{-1}$), $\theta_s$ (cm$^3$ cm$^{-3}$), $\theta_r$ (cm$^3$ cm$^{-3}$), $\alpha$ (cm$^{-1}$) and n, and all of them were affected for each of the four depths. For the automatic inversion, the WAVE model was coupled to the Shuffled Complex Evolution Metropolis algorithm for optimization and uncertainty assessment of hydrological model parameters (SCEM-UA, Vrugt et al. (2003)).



This global optimization algorithm is a Bayesian method based on the Markov chain Monte Carlo method (Gilks et al., 1998) that uses the Metropolis Hastings strategy (Metropolis et al., 1953) to evolve the population of possible parameter sets. The method reaches both the most likely parameter set and its underlying posterior probability distribution, conditioned to observed soil water data, within a single optimization run. The fit of the simulations to the observed data was evaluated by the coefficient

of efficiency (Ceff; Nash and Sutcliffe (1970)) and the root mean squared error (RMSE), as proposed by Ritter and Munoz-Carpena (2013). This process was repeated for both treatments and for each of the individual years in order to assess the dynamic evolution of the soil hydraulic properties over time. Newer indexes as Willmott et al. (2012) could have be used, but the Ceff was chosen for a better comparison with the previous work done in the experiment (Gabriel et al., 2012). Analyses of variance (ANOVA) were performed for each variable over the entire experiment, considering the treatment and year as fixed

factors. Means were separated by Duncan's multiple range test, and the statistical significance was evaluated at P ≤ 0.05.

## 3 Results and discussion

### 3.1 Weather conditions and crop development

The weather conditions for the field trial are illustrated in Figure 1. The weather conditions during the cover cropping period changed from year to year, presenting different conditions for barley development and allowing a test of the model in a broad

range of environmental conditions. The weather conditions varied from very humid seasons (e.g., 2009/10 had 612 mm of rainfall accumulated between October and April) to very dry seasons (e.g., 2011/12 had just 124 mm of rainfall in the same period). Moreover, there were also differences in the cumulative rainfall during late autumn, winter or the beginning of spring. Usually, the winter is dry, as is typically observed in Mediterranean climates, but the autumn and spring are very variable, with years in which one, both or none of them dry or wet. Temperature conditions were also different between years. Even when

the soil never got frost, there were winters in which the minimum temperature average was below 0 °C during three consecutive months (i.e., 2011/12) and others where it was always above 0 °C (i.e., 2009/10 or 2015/16). Both rainfall and temperature distributions affected the soil water dynamics and soil water balance terms. The 10-year weather series considered in this study represents the diversity of weather situations that may occur under these Mediterranean conditions.





**Figure 1: Monthly climatic conditions during the ten simulated periods. Tmax and Tmin are the absolute maximum and minimum temperature, respectively, observed within a month. Tmax_avg and Tmin_avg are, the monthly average of the daily absolute maximum and minimum temperatures, respectively.**

5   The results of the crop growth modelling of the field experiment are illustrated in Figure 2. The final observed CC biomass production in dry matter (d.m.) varied from 1145 to 5117 kg d.m. ha[-1], and WAVE was able to predict this crop development with an $R^2$=0.67 and a root mean squared error (RMSE)=1382 kg d.m. ha[-1], whereas the average observed standard error was 928 kg d.m. ha[-1]. The predicted ground cover throughout the 10 years at different crop dates after sowing matched the observed ground cover with an $R^2$=0.74 and an RMSE=16%. Predicted root depth adjusted with observations had an $R^2$=0.71 and an

10   RMSE=15.8 cm. Crop simulations were always within the range of variation observed in the field, with RMSE values similar to the natural field variability. The relative RMSEs were 24, 35 and 33% for biomass, ground cover and root depth, respectively, similar or smaller to the 35% presented by Coucheney et al. (2015) as acceptable when evaluating the crop components of an integrated agro-hydrological model. The correct simulation of these three variables in different growing





conditions is a key factor in order to achieve an accurate estimation of the evapotranspiration. And this consideration is possible because CC biomass was well correlated with total transpiration (Ritchie and Johnson, 1990), ground cover with direct soil evaporation and rainfall water interception (Tanner and Jury, 1976) and root depth with the total soil water availability for the plant (Doorenbos and Kassam, 1979).

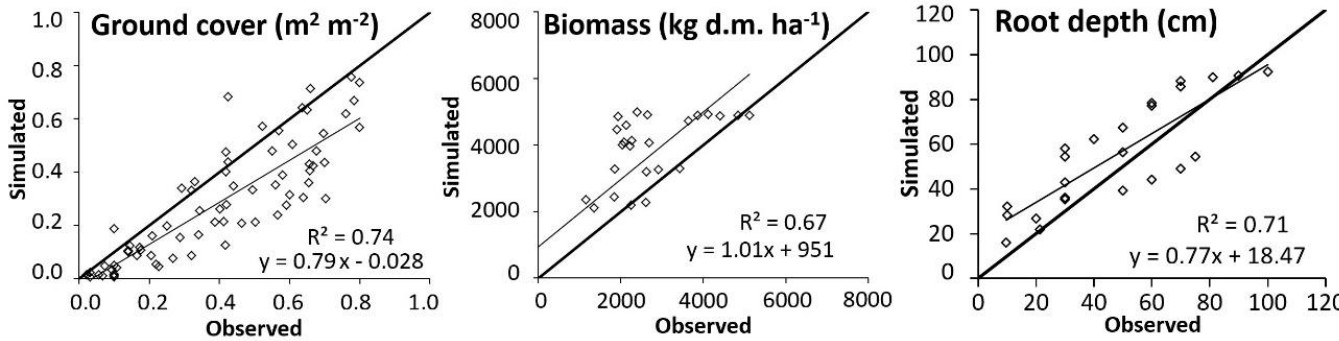

**Figure 2: Simulated versus observed ground cover, aerial biomass and root depth in the field.**

### 3.2 Soil hydraulic parameter evolution

The model adjusted the soil water content with a general coefficient of efficiency (Ceff) equal to 0.79 for the fallow and 0.83 for the barley treatment, combining all years and depths. Considering individual years, the Ceff ranged from 0.77 to 0.57 in the fallow treatment and from 0.81 to 0.64 in the barley treatment. Analysing layer by layer, the best adjustment in the fallow was obtained in the 0-20 cm soil layer, with a Ceff = 0.81, and the worst was in the 40-80 cm soil layer, with a Ceff = 0.64. In the barley treatment, the best adjustment was in the 20-40 cm layer, with a Ceff = 0.84, and the worst was in the 0-20 layer,

with a Ceff = 0.72. The Ceff obtained for all years and treatments were in the same range as the one obtained by Gabriel et al. (2012) for the same soil during 2006/07.

    The time course of the soil hydraulic properties presented two different phases (Figure 3). In the first phase, there was a clear effect on hydraulic properties at the beginning of the study (from one to three years), mainly driven by the conversion of the field site from standard reference tillage towards minimum tillage. During these years, CC and bare soil treatments changed

(or not changed) in the same way and mostly in the upper layers, the ones with a previous higher ploughing activity. There was a reduction in the residual soil water content ($\theta_r$) at 20-40 cm, in the saturation soil water content ($\theta_s$) at 0-20 and 40-80 cm and in the soil hydraulic conductivity ($K_s$) at 20-40 cm, suggesting soil compaction. A similar process was observed by other authors, such as Rücknagel et al. (2016), who reported the same compaction in CC and fallow treatments during the first years of a minimum tillage experiment. In the second phase, after two to four years, depending on the parameter and the depth,

the evolution of soil parameters changed, starting to demarcate differences between CC and the bare soil treatment.





**Figure 3: Evolution of optimized soil parameters at different depths and throughout the ten studied years. $\theta_r$ is the residual soil water content, $\theta_s$ is the saturation soil water content, $\alpha$ and $n$ are the van Genuchten curve-shape hydraulic parameters and $K_s$ is the soil hydraulic conductivity. * Symbol represents statistical differences between treatments within a year ($p < 0.05$).**



The effect on $\theta_r$ throughout the ten years was reduced to a constant increment from the first year to the fifth at 0-20 cm. This increment was probably induced by the minimum tillage (just 5 cm deep once per year) in a layer that used to be intensively tilled. There was also a recovery of the initial values of $\theta_r$ at 20-40 cm after the initial decrease. In this case, the CC favoured the increase, recovering at the third year, whereas bare soil needed 3 extra years. The effect of CCs on the time course of $\theta_r$ can be explained by the effects of roots on the development of micropore structure. The root effect on micropore structure development has been reported as being due to the micro-fissuring produced by the wetting-drying process, enhanced by the presence of roots but also by the radial pressures exerted by the roots themselves (Scanlan, 2009;Bodner et al., 2014). As barley has relatively thin roots, the effect of the CC on the micropore structure development is not very obvious, except in the upper layers, where barley root density was very high. This suggests small differences in the micropore structure between fallow and CC treatments $\theta_r$. This suggests small differences in the micropore structure between fallow and CC treatments.

The effect of CCs on the time course of $\theta_s$ was more significant compared with the effect on the time course of $\theta_r$. There were no obvious differences between treatments for the 0-20 cm depth, but some trends were observed at 20-40 cm. This trend became significant in the 40-80 cm and 80-120 cm soil layers. In this case, the $\theta_s$ of the CC treatment increased by an average of 0.06 cm$^3$ cm$^{-3}$ for the 40-80 cm soil layer and by 0.05 for the 80-120 cm soil layer since 2008/09. These results suggested an increase in macroporosity produced by biopores from dead roots and by improvements in soil structure. This result is consistent with observations from other authors who reported that macro-porosity could increase in cover cropping systems (Cresswell and Kirkegaard, 1995; Bodner et al., 2014; Yu et al., 2016). Moreover, some authors have reported that CCs increased organic matter and aggregate stability (Six et al., 2006; Peregrina et al., 2010), providing more stability to this structural improvement. In our study, this difference was most pronounced in deeper soil layers because irrigated maize roots tend to homogenize the macroporosity of both treatments in the upper soil layers.

Van Genuchten parameters ($\alpha$ and n) followed inverse trends. The $\alpha$ value was variable, but, in general, the bare soil treatment tended to decrease this value with respect to the initial conditions and the CC treatment. The CC treatment presented more stable values and even tended to increase with respect to the initial conditions throughout the soil profile. Some inverse correlation ($R^2$=0.76) has been observed in the surface layer between $\alpha$ and rainfall during the autumn months in the barley treatment, but not in the fallow. This correlation is difficult to explain, but it could be the effect of a larger particle transport (clay transport) during the wetter periods, resulting on a more clogged soil with a smaller $\alpha$. It is also important to notice that the $\alpha$ in one of the most uncertain parameters, due to its low effect on the results as was also observed in the sensitivity analysis. The n value was less variable. In this case, the CC treatment tended to decrease the n value with respect to the initial conditions and the bare soil treatment, becoming significant during the last 3-4 years, depending on the layer. An increase in $\alpha$ and a decrease in n in the CC treatment were consistent with the porosity development in this treatment. Indeed, when $\alpha$ increases, the air entry value decreases which occurred when the new macro-porosity was developed in the soil. Additionally, n measures the distribution of pore sizes. When n increases, the variation in pore sizes decreases, suggesting that some pore size classes became dominant. Both effects, together with the impacts of CC on $\theta_s$ and $\theta_r$, suggest considerable differences in the time course of the water retention curve between treatments. In the two upper layers, the CC soil water retention curves were slightly



risen and flattened, which was a result of soil with larger macroporosity development, increasing water infiltration to deeper layers and reducing the crop available water in some years (calculated as the difference between field capacity and wilting point soil water content). In our field experiment, García-González et al. (2016) and García-González (2018) measured an increase of water-stable aggregates in the CC treatment at different dates in the upper layer, supporting our results. Muñoz-

Carpena et al. (2008) also showed a similar effect on the van Genuchten hydraulic parameters in a Sunn hemp (*Crotalaria juncea* L.) cover cropping experiment, suggesting that an increase in organic matter improved soil aggregation. However, below 40 cm in depth, the crop available water (as the difference between soil water content at field capacity and permanent wilting point, 33 and 1500 kPa respectively) increased by an average of 20% from the third to the last year. This effect, together with the faster infiltration from the upper layers, could result in larger crop available water, not only in the entire profile but

also in the upper 80 cm, where the main crop explores more efficiently. Even when there were not many available results on the direct effect of CCs on soil water retention curves, Palese et al. (2014) also reported an increment in soil water retention in a cover cropped olive orchard in the deeper layers. However, they concluded that this increment was only the result of better infiltration, resulting in a reduction of water loss. Therefore, a CC can be a competitor with the main crop for water, but the improvement in soil water retention in the intermediate zone of the soil profile could minimize this competition or even turn

it into an advantage in arid and semiarid regions.

Finally, the CC treatment was also able to enhance the soil hydraulic conductivity in the layers with lower $K_s$. This effect was first observed in 2008/09 at 20-40 cm and was maintained until the end. At 80-120 cm in depth, the effect appeared later, in 2012/13 but was also persistent in time. In our field experiment, García-González (2018) measured an increase of water infiltration rates in the CC treatment at different dates in the upper layer, supporting our results. Similar increases in infiltration

rates in the upper layers of CC soil were observed by other authors (Palese et al., 2014; Yu et al., 2016). Moreover, Gish and Jury (1983) concluded that the pores generated by roots present high connectivity and facilitate water transport through the soil. Again, CC root development and improvement in the structure and pore size distribution seem to be responsible for this increase, reducing the runoff risk and the erosion problems even under the gentle slopes commonly found in semiarid and arid regions.

Some other hydraulic parameters, such as the tortuosity or hysteretic parameters, could also be included in the inversion analysis, however they were not considered to be strongly affected by the CC treatment in this study based on Kool and Parker (1987). We also considered that the CC treatment would not influence the hysteresis effects and therefore it was ignored in the analysis. We, therefore, decided to fix all these parameters to reduce identification problems in the inverse analysis. However, additional hydraulic properties and hysteresis effects could be included in future research on the effects of CCs on soil hydraulic

properties. Moreover, while inverse modelling allows inferring complex system parameters from ready available observations of dynamic system state variables, it is possible that the obtained parameter values suffer from equifinality. We reduced this equifinality problem by using different and independent parameter chains in the Markov chain methodology and by constraining the confidence interval of each parameter. We based these constraints on the initial sensitivity analysis of the model, as presented by Gabriel et al. (2012).



### 3.3 Water balance analysis

The CC treatment did not increase the simulated total water losses compared to the fallow treatment throughout the 10 years studied (Figure 4). The CC treatment increased the simulated evapotranspiration with respect to bare soil (235 vs. 162 mm year[-1] on average, respectively). However, this increase was compensated by a decrease in the simulated water drainage below

5   120 cm in depth. The bare soil simulated drainage was on average 60.7 mm year[-1] larger than the drainage of the CC treatment, but this difference reached 188 mm in very rainy years as 2009/10. These results support previous research on CCs that concluded that good management of the CC would not lead to water competition with the subsequent cash crop even during dry years as 2007 and 2011 (Clark et al., 1997; Alonso-Ayuso et al., 2014). Moreover, this result supported the theory that CCs not only reduce leaching of nitrate and other solutes but also reduce intensive drainage (Thorup-Kristensen et al., 2003).

10  Therefore, CCs can be considered a useful tool for enhancing soil water availability and environmental services linked to agricultural soils, such as the protection of groundwater from nitrate and pesticide pollution from agricultural sources. However, in years when rainfall does not occur between the CC termination date and the cash crop seeding, there might arise a problem of water competition if we only take seedbed water content into account (Clark et al., 2007).

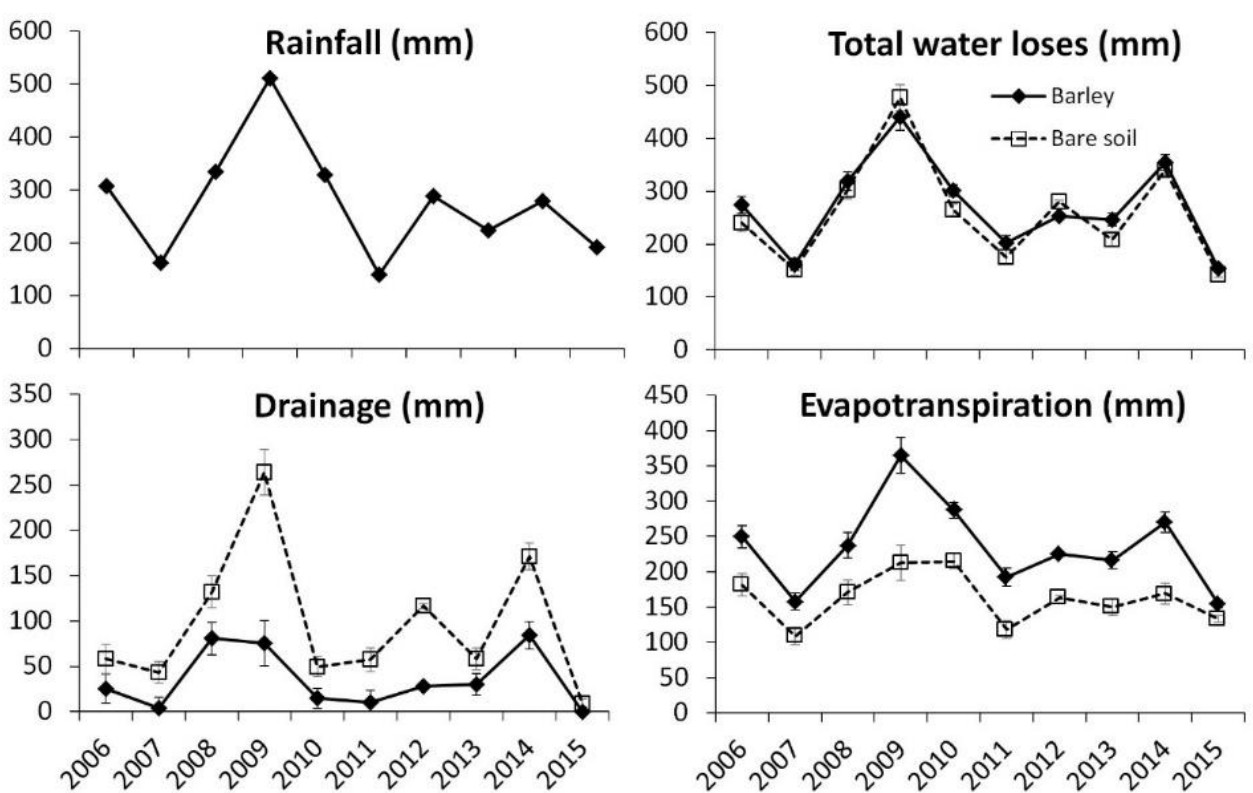

**Figure 4: Soil water balance evolution with and without a cover crop in the 120-cm depth of the soil profile and throughout the ten studied years.**




## 4 Conclusions

Using data collected in a 10-year field trial of an agricultural rotation including a CC versus fallow treatment between cash crops, we were able to elucidate by inverse modelling the impact of CCs on the medium-term evolution of the soil hydraulic properties. The cover crops were demonstrated to be a useful tool for improving the soil hydraulic functions of the agricultural

system. This improvement could be principally based on a more compensated distribution among macro- to micropores, reducing soil compaction and increasing soil water retention and the water available for the crop. Moreover, the resulting soil could be less prone to runoff and drainage losses, compensating (and even reversing) the possible water competition of the cover crop with the subsequent cash crop. This fact has special relevance in semiarid regions, where water is the most limiting factor in agricultural production.

*Competing interests*. The authors declare that they have no conflict of interest.

*Acknowledgements*. This work was supported by The Comisión Interministerial de Ciencia y Tecnología (projects AGL2011-24732 and IJCI201420175), Comunidad de Madrid (project AGRISOST, S2013/AB1-2717) and Belgium FSR 2012

cofounded by Marie Curie actions (ref. SPER/DST/340-1120525).

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
