# Peer review of "Assessing the cover crop effect on soil hydraulic properties by inverse modelling in a 10-year field trial"

_Hydrology and Earth System Sciences, 2018_

## Referee Comment (RC1) · Anonymous Referee #1 · 26 Jul 2018

The manuscript is dedicated to the estimation of the influences of a cover crop treatment on some physical properties of the soil in a 10-year field experiment. The research subject is very interesting to assess the convenience of a soil and water conservation practice in a semiarid area.

**Overall quality**

Nevertheless, the manuscript contains some problems that require a thorough revision. The authors, instead of exploring their field data, apparently prefer to extract some information of the soil water retention curve and of the saturated hydraulic conductivity through inverse methods fitting a hydrological model coupled to those data.
I think that the quality of the raw data is always greater than the use of the output of any model, in particular if the this model systematically predicts a lower ground cover or greater biomass than the corresponding observed values, as Figure 2 indicates.
There are other problems with the interpretation of soil physical parameters as indicated below.

**Specific comments**

The results shown in Figure 2 can give an 'acceptable' fit, (line, L, 12, page, P, 5), but the data points of the figure are either below, case of ground cover, or above, case of biomass in the two fist plots of the figure. This trend could affect the results.
The time variation of the optimized values of the soil physical parameters shown in Figure 3, can be related to some environmental conditions. As indicated in a previous review this was the case of the van Genuchten soil water retention equation parameter $\alpha$, in the surface layer, 0-20 cm, for the barley plots. This parameter normalizes the matric component of soil water potential, $\psi$, in brief matric potential. Using the information of figures 3 and 4, I have plotted in figure 1, below, the optimized value of the parameter $\alpha$ against the annual rainfall, $P$. The decreasing trend is evident. Why the parameter $\alpha$ can change with the annual rainfall? In principle the parameters of the soil water retention equation do not depend on the rainfall, but the observed relationship of the $\alpha$ parameter with the annual rainfall is more evident than its relationship with the time. The explanation of the manuscript, 'large particle transport (clay transport)' within the soil (L 25-26, P10) is doubtful. Could it be an artifact of the optimization method?

[Figure]

Fig. 1. Relationship between the measured annual rainfall and the optimized value of the van Genuchten parameter $\alpha$ of the soil water retention curve, for the top layer of the barley treatment.

The $\alpha$ parameter is linked to the other parameters $m$ and $n$, in the original equation of van Genuchten (1980), relating the saturation degree of water in the soil, $S$, for any water content, $\theta$, with the residual,

and saturated water content, $\theta_r$, and $\theta_s$, respectively,

$$S = \frac{\theta - \theta_r}{\theta_s - \theta_r} \tag{1}$$

with the matric potential $\psi$

$$S = [1 + (\alpha\psi)^n]^{-m} \tag{2}$$

Van Genuchten (1980) suggested a relation between the $m$ and $n$ parameters to get a closed form equation in the relation between the hydraulic conductivity of the soil and the degree of saturation

$$m = 1 - \frac{1}{n} \tag{3}$$

Kosugi[1] found a further relationship between the parameters $\alpha$, $m$ and $n$ for the inflection point of the water retention curve, which, with the help of equation 3, characterizes the value of the matric potential at this point, $\psi_{IP}$, as

$$\psi_{IP} = \frac{1}{\alpha} m^{1-m} \equiv \frac{1}{\alpha}\left(1 - \frac{1}{n}\right)^{1/n} \tag{4}$$

The equation 4 can explain the observation of L 29-32 P 10 of the manuscript.

Why the water balance analysis of the section 3.3 has not been based on the raw data, as, for instance, Palese et al (2014) did? The results of the Figure 4 induce several questions not answered in the text.

[Figure]

Fig. 2. Relationship between the measured annual rainfall and the optimized value of the evapotranspiration of the barley treatment.

The evolution of the simulated drainage volume of both treatments is roughly parallel, with greater volumes lost from the bare soil treatment than from the barley one. However, during the year 2009-2010, a great difference of drained volume, more than 150 mm, between both treatments was observed: the barley treatment lost less water by drainage than the bare soil treatement. Why? Examining the simulated evapotranspiration plot, one could think that the not drained water was evapotranspirated by the barley crop. If this hypothesis correct?

The manuscript does not inform on the slope of the ground. Was any runoff observed in the plots? Another aspect that deserves some attention is the proportionality between the measured annual rainfall and the simulated evapotranspiration from the barley treatment, very patent in a simple visual inspection of the plots, and more clearly shown in the Figure 2 above. Is there any reason for the apparent proportionality between the annual values of precipitation and simulated evapotranspiration?

With respect to the method of estimation of the evaporation from bare soil, the manuscript indicates that it was computed by the 'multiplication of the reference evapotranspiration by 1' (L18 P5). If I am interpreting this indication correctly it implies that the soil was losing water to the atmosphere without any internal restriction, the stage I of soil water evaporation[2]. Is this interpretation correct? In the affirmative case, why?

**Technical comments**

Certain terms in the manuscript are imprecise, as 'water availability' written in L23 P1 and in L2 P11, and not defined until L 8 P11.

The field capacity was estimated following Assouline and Or (2014) suggestions in L13 P3, but this estimation seems forgotten in L8 P11, when the value 33 kPa is adopted without any further explanation. Why?

What was the purpose of the measurement of saturated hydraulic conductivity in the laboratory of indicated in L9 P4?

What air entry values were found by the authors (L 30-31 P10)?

The manuscript needs a revision to repair some formal defects. For instance, the sentence in L9 P10 is repeated in L10 P10. Some sentences reiterate certain terms like 'parameters' in L 29-35 P2. The sentence in L 22-23 P 6 is obvious.

**Additional references**

[1]   Kosugi, K. 1994. Three-parameter lognormal distribution model for soil water retention. *Water Resour. Res.* 30:891-901.

[2]   Or, D., Lehmann, P., Shahraeeni, E., Shokri, N. 2013. Advances in soil evaporation physics. A review. *Vadose Zone J.* doi:10.2136/vzj2012.0163.

---

## Referee Comment (RC2) · Anonymous Referee #2 · 14 Aug 2018

Review of "Assessing the cover crop effect on soil hydraulic properties by inverse modelling in a 10-year field trial" submitted to HESS by Jose Luis Gabriel, Miguel Quemada, Diana Martín-Lammerding, and Marnik Vanclooster

Gabriel and co-workers present a very comprehensive study on the effect of barley cover-cropping and minimum tillage on the hydraulic soil properties based on a 10-year experiment and subsequent inverse modelling for parameter identification. The study and the manuscript are well thought and worked out. It definitely fits in the scope of HESS and will be a fruitful contribution to the field.

That said and with all respect to this work, I however see some fundamental issues

with the presented study which I would suggest the authors to reconsider:

1. Why is there no direct soil data from some time within the 10-year experiment used for validation?

2. How can one account for the effect of changing soil conditions on the initially calibrated sensor readings?

3. How do the authors account for equifinality and parameter interaction in their inverse modelling?

4. Is the WAVE-WOFOST model system evaluated for its general necessity/suitability in the study and potential parameter interaction and identifiability of the required soil parameters?

**(1 & 2)** The authors base their study on continuous measurements of soil moisture in 6 depth levels (0.1 to 1.1 m, 0.2 m increments) on six treatment plots. In addition meteorological state variables are recorded on site. As very good practice, the soil moisture sensors have been calibrated under field and lab conditions to the specific soil condition previous to the experiment period. At this time a pedohydrological analysis was done, too. Since the study is about the effect of cover crop and tilling routines on soil hydraulic properties, I suspect that the authors have considered to directly sample the soil repeatedly to answer their research question.

What I find especially challenging is that a calibration of the capacitive sensors to a specific soil condition might be in question once the soil matrix properties change. As such, there might be an issue that the changing soil properties under study may have indiscernible effect on the actual measured dynamics. With regard to interaggregate and macropore structure formation, the effect should be more clear to be identified as the resulting dynamics do not rely on precisely measure the absolute soil moisture

but the relative dynamics during events. However, the authors do not show direct pedophysical measurements from during the experiments, nor do they discern different (event-scale) soil moisture dynamics (diffusive redistribution vs. initially advective percolation in soil macro structures). I find that an unnecessary flaw in the well-thought study. With regard to the presented results, I suspect that a more detailed view to different temporal integrals of the hydrological responses might allow for more specific and less equifinal parameter identification.

**(3)** The authors present a multi-step parameterisation scheme, which first addresses the crop parameters based on observed properties like biomass production, ground cover and root development. In a second step they used the Shuffled Complex Evolution Metropolis algorithm for optimisation (and uncertainty assessment) of the pedohydrological parameters. While this appears as very reasonable choice to derive the parameters for the system, it still cannot resolve the issue of potential parameter interaction and equifinality. Especially so, when the objective function evaluates the full observation time series and does not include the crop parameters. Given the criterions (NSE and RMSE), I would expect the parameters to trend towards acceptable timing and fit of the mean soil moisture. This to my understanding would not be sensitive to changes in bulk density but might only weakly consider the creation of macro-structure (which the authors point out to be one important process). A Richards-type single-domain model might still be capable to assess the research questions, but I suspect that some event responses and respective percolation properties, should be studied in more detail. For this, a more data-focussed approach might be the first step.

**(4)** Assumingly in order to dynamically determine evaporation, the authors coupled the Richards-type soil water model WAVE with the crop model WOFOST. While this appears to be a very reasonable step, it might not be central for the event-scale soil water dynamics. Moreover, it introduces a quite large number of potentially interacting parameters. Even if one reduces the about 120 crop parameters to the most sensitive 10, their interaction with the soil parameters is still in question. From the biomass plot in fig.

2 I get the impression that the model actually only simulates 3 to 4 different "classes" of production, which are not really coherent with the observations (maybe except for the very lowest ones). This actually is in line with my expectations that WOFOST (unfortunately) is not really capable to be used in a dynamic eco-hydrological setup. For the study at hand, erroneous crop simulations may be susceptible to blur the actual soil parameter effects without providing the desired advances of more correct evaporation simulations. Since apparently ground cover of the cover crops has been observed occasionally and since WOFOST suggests only 3-4 crop development scenarios, maybe a more direct implementation of an ET estimate might reduce ambiguity in the analysis?

Despite my concerns and suggestions for the methods used in the study, with regard to the current setup I find it necessary that the authors give more insight into the observed and modelled soil water dynamics, the parameters apart from fig. 3, the used time stepping, and at best some more details about the actual model realisations (especially since the authors use their own Matlab derivatives). This could partly also be given as supplement.

As such, I suggest the manuscript to be considered for major revisions.

**Minor comments:**

**P2L4:** soil size pore distribution » pore size distribution?

**P2L31f:** I am not sure, if inverse modelling is specifically useful "to overcome a parameter limitation problem" as it faces the issue of parameter interaction and equifinality. I would expect more specific explanation and citations here.

**P3L2:** Although I agree to the general attitude that "multi-sensor" probes have advantages, I do not see that the study could not be done with more standard soil moisture probes. Especially with regard to the nature of capacitive sensors being potentially more effected by changes in the soil properties, one could also think of alternative

setups - eg. using TDR probes.

**P3L3ff:** I would expect that the uncertainties are not directly depending on the identification strategy of the parameters. Thus a consistent measurement over depth might allow for the assumption that uncertainties between the individual records might be reduced. However, under natural conditions there might always be air and gravel entrapments altering the control volume. Maybe I misunderstood the statement?

**P3L18:** I would expect the weather station's sensors to be more important than the logger...

**P3L24f:** I can grasp the study layout from the description. However, I would suggest a small plot, clarifying on the locations of the random plots, the respective treatments and the locations of the observation stations.

**P4L2:** I do not quite understand: There are 8 randomly chosen plots, but only in 6 soil moisture was monitored?

**P4L4f:** This calibration is very good practice. However, your experiment might raise the question if such a calibration remains valid for changing soil conditions... I expect this also holds true for the soil hydraulic properties in general. I would suggest to include a paragraph on this in the discussion. Section 2.3: The model system appears very parameter-rich and finally rather complex. Although I can follow your description having once coupled WOFOST with the hydrological model SWAP, I am not convinced that this description suffices to be able to understand the coupled model system and to reproduce your results. Moreover at this stage of reading, I slightly doubt that the model system is actually required to answer the research question.

**P4L28f:** Was WOFOST used to determine the crop development and soil water use? How has it been integrated? From my experiences, coupling WOFOST and any hydrological model may result in even worse identifiable parameter sets since the crop parameters compensate for soil definitions and vice versa.

**P6L15:** I would not consider 612 mm as "very humid" » relatively humid?

For the rest of the paper, I refer to my general remarks above.

---

## Author Comment (AC1) · 5 Sep 2018

Thank you for the review. Here you can find attached the response and the new version of the manuscript including your suggestions. Best regards, The authors.

Please also note the supplement to this comment:
https://www.hydrol-earth-syst-sci-discuss.net/hess-2018-372/hess-2018-372-AC1-supplement.zip